# The Large Molecular Weight Polysaccharide from Wild Cordyceps and Its Antitumor Activity on H22 Tumor-Bearing Mice

**DOI:** 10.3390/molecules28083351

**Published:** 2023-04-10

**Authors:** Li Tan, Sijing Liu, Xiaoxing Li, Jing He, Liying He, Yang Li, Caixia Yang, Yong Li, Yanan Hua, Jinlin Guo

**Affiliations:** 1Key Laboratory of Characteristic Chinese Medicine Resources in Southwest China, College of Pharmacy, Chengdu University of Traditional Chinese Medicine, Chengdu 611137, China; 2College of Medical Technology, Chengdu University of Traditional Chinese Medicine, Chengdu 611137, China; 3School of Public Health, Chengdu University of Traditional Chinese Medicine, Chengdu 611137, China

**Keywords:** polysaccharide, wild *Cordyceps*, structural characterization, anti-tumor activity, apoptosis

## Abstract

Cordyceps has anti-cancer effects; however, the bioactive substance and its effect are still unclear. Polysaccharides extracted from *Cordyceps sinensis*, the fugus of Cordyceps, have been reported to have anti-cancer properties. Thus, we speculated that polysaccharides might be the key anti-tumor active ingredients of Cordyceps because of their larger molecular weight than that of polysaccharides in *Cordyceps sinensis*. In this study, we aimed to investigate the effects of wild Cordyceps polysaccharides on H22 liver cancer and the underlying mechanism. The structural characteristics of the polysaccharides of WCP were analyzed by high-performance liquid chromatography, high-performance gel-permeation chromatography, Fourier transform infrared spectrophotometry, and scanning electron microscopy. Additionally, H22 tumor-bearing BALB/c mice were used to explore the anti-tumor effect of WCP (100 and 300 mg/kg/d). The mechanism by WCP inhibited H22 tumors was uncovered by the TUNEL assay, flow cytometry, hematoxylin–eosin staining, quantitative reverse transcription–polymerase chain reaction, and Western blotting. Here, our results showed that WCP presented high purity with an average molecular weight of 2.1 × 10^6^ Da and 2.19 × 10^4^ Da. WCP was determined to be composed of mannose, glucose, and galactose. Notably, WCP could inhibit the proliferation of H22 tumors not only by improving immune function, but also by promoting the apoptosis of tumor cells, likely through the IL-10/STAT3/Bcl2 and Cyto-c/Caspase8/3 signaling pathways, in H22 tumor-bearing mice. Particularly, WCP had essentially no side effects compared to 5-FU, a common drug used in the treatment of liver cancer. In conclusion, WCP could be a potential anti-tumor product with strong regulatory effects in H22 liver cancer.

## 1. Introduction

As one of the most malignant tumors in the world, liver cancer imposes a huge burden on people. According to the International Agency for Research on Cancer (IARC) 2020 estimates, liver cancer is the sixth most commonly diagnosed cancer and the third leading cause of cancer death worldwide [1]. Liver cancer is also the second leading cause of premature death from cancer [2]. China alone accounts for 45.3% of global liver cancer cases and 47.1% of liver cancer deaths [3]. Liver cancer can be divided into two major categories, primary hepatocellular carcinomaoriginate (PHC) and secondary liver cancer, the latter of which is known as sarcoma and is less common than PHC. PHC originates in the epithelial or mesenchymal tissue of the liver and is a highly prevalent and extremely dangerous malignancy in China [4]. Hepatocellular carcinoma (HCC) is its main pathological type, accounting for approximately 85–90% of PH [5,6]. HCC is the fourth most common malignant tumor and the second leading cause of tumor-related death in China, posing a major health threat to the Chinese people. Hepatectomy and liver transplantation are the mainstream treatment options for HCC. However, for most patients with advanced HCC, the disadvantages of surgery usually outweigh the advantages due to late diagnosis. More importantly, patients with HCC may experience serious adverse effects and drug resistance during chemotherapy, resulting in a far-from-satisfactory prognosis for HCC [7]. Therefore, it is important to explore natural anti-tumor agents with potent tumor-suppressing abilities and minimal side effects. In recent years, traditional Chinese medicine (TCM) treatment has received a lot of attention from researchers due to its ability to improve clinical symptoms, enhance body resistance, reduce the adverse effects of radiotherapy and chemotherapy, and improve the quality of life of patients [8].

Polysaccharides are biologically active macromolecules that contain 10 or more monomers linked by glycosidic bonds. In recent years, natural polysaccharides have attracted the attention of researchers due to their safety and biological activity [9]. Studies have reported that polysaccharides extracted from TCM have few adverse effects and various biological functions on cancer [10], hyperlipidemia [11], inflammation [12], immunity [13], and antioxidation [14]. Polysaccharides from *Penthorum chinense* Pursh have been shown to inhibit H22 cell growth via the induction of a mitochondrial-dependent apoptosis [15]. A water-soluble polysaccharide from *Eucommia folium* exhibited more potent toxicity on H22 cells [16]. Liu et al. reported that an alcohol-soluble polysaccharide from *Bletilla striata* can significantly inhibit the growth of H22 cells at a certain concentration [17]. A homogeneous polysaccharide (PHP-1) obtained from *Pseudostellaria heterophylla* can exert anti-tumor effects in H22 tumor-bearing mice [18]. In addition, Chen et al. determined that polysaccharides from *Taraxacum mongolicum* leaf modulated the p53 signaling pathway to inhibit H22 tumor growth [19]. The anti-tumor capacity of an acidic water-soluble polysaccharide from *Bupleurum chinense DC* is mainly achieved through the induction of apoptosis and arresting of the S phase of H22 cells [20].

Cordyceps is a complex of ascospores formed by the *Cordyceps sinensis* (Berk.) Sacc. (*C. sinensis*) parasite that is found on insect larvae from the Hepialidae family and the bodies of the larvae. It is mainly distributed in high-altitude areas around 4000 m above sea level in Sichuan, Yunnan, Tibet, Qinghai, China [21]. Studies have reported that Cordyceps consists of various chemical constituents, such as nucleosides, lipids, saccharides, mannitol, and amino acids [22]. Modern research has shown that Cordyceps possesses many bioactive functions, such as anti-tumor [23], anti-inflammatory [24], antioxidation [25] and immunomodulation [26] activities. However, its anti-cancer composition is unclear. *Cordyceps sinensis* is fermented from microbes isolated from Cordyceps [27] and its polysaccharides have anti-cancer properties against liver cancer [28], melanoma [29], and colon cancer [30]. Thus, we speculated that polysaccharides are the key active ingredients of Cordyceps against cancer. The molecular weight of *Cordyceps sinensis* polysaccharide is significantly different from that of Cordyceps, which has a larger molecular weight [31]. Moreover, some studies have shown that larger-molecular-weight polysaccharides have better anti-cancer activity [32]. The above studies suggested that Cordyceps polysaccharides may have tumor-inhibiting efficacy. However, the anti-tumor activity and underlying mechanism in vivo are rarely reported.

In this study, we sought to analyze the structure of Cordyceps polysaccharides and their anti-liver cancer function assessed in vivo using an H22 mouse model. We also aimed to explore its potential mechanisms, which may provide valuable information for the development of natural anti-tumor drugs.

## 2. Results

### 2.1. The Total Sugar Content, Purity, Average Molecular Weight, and Microstructural Feature Results of WCP

The yield of polysaccharide was 3.2% by water extraction and the content of total sugar in WCP was about 72.9% according to the standard curve (Figure 1A). In the UV spectrum of WCP, there were no obvious absorption peaks at 260 nm and 280 nm (Figure 1B), suggesting that WCP was free of proteins and nucleic acids. Consistent with this, the results of SDS-PAGE gel electrophoresis also showed no obvious protein in WCP (Figure 1C). The average molecular weight of WCP was estimated by HPGPC. According to Figure 1D, WCP had three distinct peaks on HPGPC, indicating predominantly three molecular weight distributions for WCP. Only two molecular weight distributions could be calculated because the peak with the largest molecular weight was outside the analytical range of the column. According to the retention time, they were 2.1 × 10^6^ Da and 2.19 × 10^4^ Da, respectively. Additionally, the surface morphology of WCP was observed by scanning electron microscopy assay. As shown in Figure 1E,F, the WCP was determined to be flocculent, with a rough surface, large pores, and a low number of cracks.

### 2.2. Main Organic Functional Groups and Monosaccharide Composition of WCP

FT-IR spectroscopy was used to detect the main organic functional groups in WCP. As shown in Figure 2A, there were two strong and broad absorption peaks at 3397.08 and 3192.77 cm^−1^, which could be derived from the O–H stretching vibrations [33]. The faint absorbance at 2942.76 cm^−1^ was suggestive of the presence of C–H. The absorption band at 1631.24 cm^−1^ was attributed to C–O asymmetric stretching vibration. The characteristic peaks at 1403.51 and 1296.47 cm^−1^ were caused by the C–H deformation vibration. These peaks are characteristic absorption peaks for polysaccharides [34]. In addition, 800–1200 cm^−1^ is the fingerprint area of carbohydrates. The signals at 1053.77 cm^−1^, 1037.33 cm^−1^, and 909.84 cm^−1^ were due to the presence of pyranose [35].

Liquid chromatography (LC) was used to determine the monosaccharide composition of WCP. As shown in Figure 2B,C, LC analysis revealed that the WCP was composed of mannose, glucose, and galactose.

### 2.3. WCP Alleviates the Symptoms of H22 Tumor-Bearing Mice

An in vivo mouse model of H22 tumors was established to assess the carcinogenic effect of WCP. Prior to inoculation with H22 hepatocellular carcinoma cells, the mice weighed similarly and there were no differences between the groups. In addition, all mice had shiny coats, good mental conditions, and normal excretions. After the model was successfully established, during the treatment period, the 5-FU group showed symptoms such as diarrhea, blood in the stool, dark urine, matted hair, and poor mental conditions, while the other groups were in good shape. Meanwhile, the body weight of mice increased steadily during the treatment, except that of mice in the 5-FU group (Figure 3A). Meanwhile, the food intake of the 5-FU group was lower than that of the WCP group at the later stage of the experiment (Figure 3B), which could be inferred that the WCP may have fewer adverse effects than 5-FU when administered to mice. At the end of the experiment, the tumor weight and volume of the High WCP groups were greatly reduced compared with the model group (Figure 3C–E, *p* < 0.01). Consistent with these results, the serum AST and ALT in the model group were significantly higher than those in the control group (Figure 3F,G, *p* < 0.05). However, this up-regulation was markedly decreased by WCP and 5-FU (Figure 3F,G, *p* < 0.05). These results indicated that WCP significantly inhibited tumor proliferation and transaminase elevation with low-toxicity side effects.

### 2.4. Analysis of the Blood Routine Results

The development and progression of tumors not only causes lesions in organs but also affects the blood composition. Therefore, we analyzed the blood routine examination by BC-5120 automatic blood analyzer. The results of H22 tumor-bearing mice in each group are shown in Table 1, including indicators relevant to leukocytes (percentages and populations of lymphocytes, percentages and populations of neutrophils), erythrocytes (erythrocyte amounts, hemoglobin, hematocrit, mean corpuscular volume, mean hemoglobin, mean hemoglobin concentration, erythrocyte distribution width coefficient, erythrocyte distribution width standard), and platelets (blood platelet, mean platelet volume, platelet distribution width, thrombocytocrit, platelet larger cell count, platelet larger cell ratio). Compared with the NC group, the model groups presented higher levels of neutrophil amounts/proportions and platelets (*p* < 0.001), and decreased levels of lymphocyte amounts/proportions, erythrocyte amounts, and hemoglobin (*p* < 0.05, *p* < 0.001, *p* < 0.01), indicating that the proliferation and invasion of cancer cells would suppress lymphocyte activity, as well as inducing anemia and inflammation effects in the host. WCP treatment was effective in improving the immunity of anti-tumor lymphocytes (*p* < 0.001) and relieving adverse effects caused by cancer. Moreover, WCP did not have a significant effect on leukocyte and blood platelet levels compared with the model group. However, when treated with 5-Fu, these H22 tumor-bearing mice showed a myelosuppression response compared to the WCP group, highlighted by leukocytopenia (*p* < 0.0001) and thrombocytopenia (*p* < 0.0001). The results of routine blood tests showed that WCP could achieve cancer suppression by enhancing the immunity of H22 tumor-bearing mice without significant myelosuppressive side effects.

### 2.5. Effect of WCP on the Liver, Lung, Organ Indices, and IL-10

The previous results of tumor weight, volume, and transaminase suggest the effectiveness of WCP in inhibiting hepatocellular carcinoma; we would like to further investigate the effect of WCP on the major organs and serum interleukin 10 (IL-10) in mice, and whether the mice have metastasis in the liver and lung. Bouins staining of lungs (Figure 4A) and lung indices (Figure 4B) showed that there were no tumor metastases in the H22 tumor-bearing mice group. HE staining of the liver (Figure 4C), liver indices (Figure 4D), and lung indices showed that Cordyceps polysaccharides were not toxic. The thymus index (Figure 4E) of the model group was significantly decreased (*p* < 0.05) compared with the NC group, suggesting that tumor cell proliferation had a harmful effect on the thymus. The thymus index increased when the dose of WCP increased from 100 to 300 mg/kg, suggesting that WCP protected the thymus from damage. When compared to the NC group, the spleen index (Figure 4F) of the model group was significantly increased (*p* < 0.01). The spleen index in the High WCP group significantly decreased compared to the model group (*p* < 0.05). The changes in immune organ indices in the WCP group indicated that WCP might have an immune-enhancing effect on H22 tumor-bearing mice. However, we also determined that the 5-FU exhibited strong toxicity to immune organs compared with the model group. Based on the results of blood tests and immune organ indices, we hypothesized that WCP could achieve anti-cancer effects by enhancing immunity in mice, so we measured IL-10 in serum, which is closely related to tumor development. As shown in Figure 4G, the IL-10 concentration in the model group was remarkably higher than that in the NC group (*p* < 0.001). Notably, the Low WCP and High WCP treatment significantly reduced the IL-10 concentrations compared to the model group (*p* < 0.0001). However, the decrease in the IL-10 in the 5-FU group was not significant. These results indicated that the tumor did not metastasize in the liver and lungs and that WCP was able to improve the body’s immune system and reduce IL-10 levels in the blood serum without toxicity.

### 2.6. WCP Changes the Proportions of T Lymphocytes and Macrophages

To further investigate the effect of WCP on the immune function of tumor-bearing mice, the distributions and proportions of T cell subsets and macrophages in the spleen were analyzed using flow cytometry. As shown in Figure 5A,B, the percentage of CD4^+^ T cells was remarkably lower in the model group compared with the NC group, suggesting that CD4^+^ T cell activity was suppressed by H22 tumors. WCP treatment increased the proportion of CD4^+^ T cells, which might be responsible for the high rate of inhibition of H22 solid tumors. When treated with 5-FU, the percentage of CD4^+^ T cells had a downward trend compared to the WCP group, which was similar to the percentage in the model group, indicating the immunosuppressive effects of 5-FU. Meanwhile, the percentage of CD8^+^ T cells was remarkably lower in the model group compared to the NC group (Figure 5A,C, *p* < 0.001). In contrast, WCP treatment could significantly increase the proportion of CD8^+^ T cells (Figure 5A,C, *p* < 0.01). The changes in the number and distribution of CD4^+^ and CD8^+^ T cells suggest that WCP inhibited tumor proliferation probably by activating cellular immunity. The large number of macrophages in the spleen has a powerful role in the phagocytosis of antigenic particles and also acts as an antigen-presenting cell (APC), regulating and enhancing the immune response. In addition to releasing tumor necrosis factor (TNF-α), IFN-γ, IL-1 and other active substances directly involved in anti-tumor effects, they also regulate the anti-tumor effects of cytotoxic T lymphocytes (CTL) and lymphokine-activated killer cells (LAK), and have a wide range of immune sensing and effector functions. Therefore, we also measured the number of macrophages in the spleen. WCP treatments could significantly increase the proportion of macrophages (Figure 5D,E). The above experimental results suggest that WCP inhibits H22 tumor proliferation probably by enhancing the immune function of mice.

### 2.7. WCP Treatment Promotes Apoptosis of Tumor Cells

The effect of WCP on the inhibition of H22 tumor proliferation was obvious, so we speculated that it may have a direct cytotoxicity effect in addition to enhancing immune function in mice. Therefore, we used hematoxylin and eosin (H and E) staining to assess the histopathological changes in the tumor. As shown in Figure 6A, the tumor cells in the model group were plump and intact. After treatment with WCP and 5-FU, the tumor cells exhibited large areas of necrosis (green arrows), which suggested that the significant anti-tumor efficacy of WCP in H22 tumor-bearing mice might involve the induction of cell death. The TUNEL method is a conventional method to detect apoptosis. Based on the results of HE staining, we used the TUNEL method to determine apoptosis in tumor tissues. The TUNEL assay results are shown in Figure 6B,C. Cells with green granules were considered to be positively stained. The percentage of positive cells in the model group was 25.35 ± 9.96. The WCP-treated groups showed an increase in the amounts of cells undergoing apoptosis, with values of 43.03 ± 11.80 and 62.27 ± 17.32 (*p <* 0.01). These results suggest that WCP inhibition of the proliferation of H22 tumors may also promote apoptosis.

### 2.8. WCP Promotes Cyto-c/Caspase8/3 and Inhibits IL-10/STAT3/Bcl2 Pathway

Given the ability of CD4/CD8 cells to regulate the downstream Caspase8 signaling pathway, we analyzed the genes and proteins associated with this pathway. As shown in Figure 7, WCP did not change the IL-6, IL-Iβ, NF-κB, and TNF-α mRNA levels. However, apoptosis-related mRNA Bax and Bcl2 were significantly changed in the WCP group. Compared with the NC group, Bax mRNA was significantly increased and Bcl2 mRNA was remarkably decreased in the WCP group (Figure 7E,F *p <* 0.05), which was consistent with the results of HE and TUNEL assay results. Therefore, we further measured the changes in apoptosis-related proteins using WB. In line with our expectations, WCP remarkably promoted Cyto-c, Caspase8, and Caspase3 expression (Figure 8A–C, *p <* 0.05; the original images are shown in Appendix A). On the other hand, the decrease in serum IL-10 suggested that WCP against hepatocellular carcinoma might occur through an IL-10-regulated signaling pathway, so we examined its downstream STAT3 and p-STAT3 proteins. As shown in Figure 8F–I (the original images are shown in Appendix A), WCP remarkably promoted Bax expression as well as the ratio of Bax/Bcl2. In addition, WCP significantly suppressed p-STAT3Y705 and Bcl2 expression in tumor tissues (*p* < 0.05). Taken together, WCP exerted its anti-tumor effects probably through the Cyto-c/Caspase8/3 and IL-10/STAT3/Bcl2 pathway in H22-tumor-bearing mice.

## 3. Discussion

Cordyceps is recorded in the Chinese Pharmacopoeia as a tonic for nourishing the kidneys, invigorating the lungs, stopping bleeding, and resolving phlegm, used to strengthen the immune system and tonify the deficient Qi [36]. The accuracy and sensitivity of UV is poor and there are many interfering substances. The addition of SDS-PAGE to the purity test is making up for the shortcomings of the UV method. In addition, polysaccharide purity can also be analyzed by cellulose acetate film electrophoresis, glass fibre paper electrophoresis, gel column chromatography, paper chromatography and agarose gel electrophoresis. As mentioned earlier, Cordyceps has anti-cancer effects, but its components of anti-tumor effect are not yet clear. The polysaccharides extracted from the *Cordyceps sinensis*, which is isolated from Cordyceps, have anti-lung, melanoma, and colon cancer effects, so it is assumed that polysaccharides are the key components of the anti-cancer effect of Cordyceps. The molecular weight of polysaccharides in *Cordyceps* is mainly 9.6 × 10^6^ Da and 2.1 × 10^6^ Da, and there is a larger molecular weight that cannot be calculated because it is outside the range of the analytical column. The molecular weight of polysaccharides in *Cordyceps sinensis* is mostly 2.8 × 10^4^ Da [31], and there is an obvious difference between them. Moreover, research has shown that larger-molecular-weight polysaccharides have better anti-cancer effects [32]. To explore the WCP anti-H22 hepatocellular carcinoma effect and elucidate the underlying molecular mechanisms, here, we present new data showing that 4 weeks of treatment with WCP in H22 tumor-bearing mice not only inhibited tumor growth but also increased levels of apoptosis. The significant inhibition of tumor proliferation and the significant down-regulation of AST and ALT levels indicated that WCP had a good effect in inhibiting liver cancer and alleviating liver damage. Furthermore, WCP did not cause weight loss or myelosuppression in animals while resisting liver cancer. In addition, the mice in the WCP group were in good spirits and had normal bowel movements, close to the condition of mice in the NC group. 5-FU is a chemotherapeutic agent that affects the synthesis of DNA and thus has an anti-cancer effect [37]. However, during the treatment period, the mice in the 5-FU group continued to lose weight, suffered from diarrhea and blood in the stool, and became depressed and progressively weaker. Even worse were the typical side effects of myelosuppression caused by 5-FU, highlighted by a significant reduction in leukocytes and blood platelets (*p* < 0.0001).

Contrary to 5-FU, WCP showed a powerful immune-enhancing effect. The thymus and spleen are the main immune organs that harbor a variety of immune cells, including T cells, granulocytes, and macrophages, and protect the body from infection and cancer [38]. The weight of these organs reflects the body’s immune capacity and is initially assessed in anti-tumor experiments [39]. In the present study, WCP significantly increased the proportion of CD4^+^ and CD8^+^ T lymphocytes as well as macrophages (*p* < 0.05) and alleviated H22-tumor-induced thymic atrophy (*p* < 0.05) and splenomegaly (*p* < 0.05). These results suggested that WCP is effective in reducing these side effects caused by tumor growth.

We further discovered massive necrosis and apoptosis of tumor cells after WCP treatment through HE and TUNEL experiments (*p* < 0.05). Increased tumor cell apoptosis inhibits cell malignancy and slows tumor proliferation [40]. Apoptosis is a complex mechanism involving two main pathways, the endogenous and exogenous pathways. The endogenous pathway is triggered by external signals received by cells that are no longer needed, which are then transmitted from the cell surface to intracellular signaling pathways, leading to the activation of Caspase8, which then promotes the activation of the effector Caspase3. Exogenous signaling pathways are triggered by stimuli that cause stress or damage to the cell. Intracellular signaling causes changes in the mitochondrial membrane, leading to the release of Cyto-c into the cytoplasm, which results in the formation of apoptotic bodies that cleave and activate the effector Caspase3 [41]. As members of the cysteine aspartate-specific proteases, Caspase8 and Caspase3 are the initiator and effector of caspase-dependent apoptosis, respectively [42]. Qi et al. [30] reported that Cordyceps sinensis polysaccharide (CSP) inhibited colon cancer cell growth by inducing apoptosis through the Caspase8/3 pathway. T lymphocytes are induced to secrete FasL, which binds to the corresponding ligand, Fas, and the Fas receptor is activated by trimerization. The activated receptor binds to FADD and then interacts with Caspase8 to activate the latter, forming a death-inducing signalling complex, which in turn activates a series of Caspase1, 3, 7, etc., promoting apoptosis in cells where Fas proteins are present [43]. Given the significant effect of WCP on CD4^+^ and CD8^+^ (*p* < 0.05), we also explored the relevant indicators of this pathway. We also discovered that WCP significantly up-regulates the expression of Caspase8 and Caspase3.

On the other hand, in this study, we also determined that IL-10 levels were significantly reduced after WCP treatment. IL-10 is an anti-inflammatory cytokine [44]. Elevated levels of IL-10 were reported to promote tumor development [45]. In addition, IL-10 is able to promote the conversion of STAT3 to p-STAT3, which enters the nucleus to regulate downstream genes related to the cell cycle and apoptosis [46]. This may also be another reason why WCP induces apoptosis in tumor cells. Therefore, we examined the levels of STAT3 and p-STAT3 protein and determined that WCP significantly reduced the expression of p-STAT3 in tumor tissue. We further examined the downstream targets of STAT3 and discovered that the expression of both the gene and protein of Bcl2 were significantly decreased. We also discovered that WCP significantly up-regulated the expression of Bax and Cyto-c, especially the ratio of Bax/Bcl2. Some studies have shown that the upregulation of Bax/Bcl2 ratio promoted apoptosis in esophageal cancer [40] as well as prostate, bladder, and renal cancer cells [47]. The ratio of the amount of Bax to Bcl-2 protein determines the ratio of homodimer to heterodimer, which plays a key role in susceptibility to apoptosis [48]. This is consistent with our results. However, this study also has some limitations. H22 cells were injected subcutaneously into the axilla of the left forelimb of mice, resulting in ectopic liver cancer. This model does not mimic the key features of the human disease process. Therefore, an orthotopic animal model should be considered to further investigate the precise mechanisms of WCP in HCC.

Based on the data from this study, a model of WCP inhibition of H22 cell proliferation was proposed (Figure 9).

## 4. Materials and Methods

### 4.1. Materials and Reagents

The wild Cordyceps were collected from Kang ding, Sichuan Province, China. H22 hepatoma cells were kindly provided by Professor Zhang Tian’e (Chengdu University of Traditional Chinese Medicine, Sichuan). Standards of monosaccharides were provided by Sigma (Saint-Louis, MO, USA). Standards of Pullulan (Narrow MWD) were purchased from Shodex, Yokohama, Japan. The 5-fluorouracil (5-FU) for injection was purchased from Abmole Bioscience (Shanghai, China). Aspartate aminotransferase (AST) and alanine aminotransferase (ALT) were purchased from Chengdu Illio Technology Co., Ltd. (Chengdu, China). Rabbit polyclonal antibodies against B-cell lymphoma-2 (Bcl-2, AF6139), Cytochrome C (Cyto-c, AF0146), and Caspase3 (AF6311) antibody were purchased from Affinity Biosciences (Nanjing, China) and BCL-2-associated X protein (Bax, BA0315-2) antibody was purchased from Boster Biological Technology Co., Ltd. (Wuhan, China). Rabbit monoclonal antibodies against Caspase8 (PTM-6085) antibody were purchased from PTM Biolab Co., Ltd. (Hangzhou, China), and signal transducer and activator of transcription3 (Stat3, BM4052), phosphorylated signal transducer and activator of transcription3 (p-STAT3 Y705, BM4835) were purchased from Boster Biological Technology Co., Ltd. (Wuhan, China). All other chemical reagents were of analytical grade.

### 4.2. Preparation of Wild Cordyceps Polysaccharide (WCP)

The dried wild Cordyceps was crushed into a fine powder. The powder samples (30 g) were defatted with 300 mL 80% ethanol at 4 °C overnight. The residue was collected by centrifugation and extracted three times with hot water (80 °C) at a ratio of 1:30 (*w*/*v*) for 2 h. The mixture was centrifuged at 5000× *g* for 10 min to remove residue. The supernatants were concentrated under reduced pressure at 60 °C to obtain an extract solution, which was precipitated by 95% ethanol (5 × the extract solution volume) and stored at 4 °C overnight [49]. Proteins were removed by the Sevag method. The precipitation was desalted using 0.05 M Tris-HCI buffer (pH 7.0) for 8 h at a time and resubmitted at least three times with fresh dialysate. Finally, the product was dried at −60 °C under a vacuum to produce polysaccharide powder [50].

### 4.3. WCP Characterization

#### 4.3.1. Total Sugar Content and SDS-PAGE Gel Electrophoresis Analysis

The content of total sugar was measured by phenol-sulfuric acid method. D-glucose was used as the standard [51]. The polysaccharide purity was first evaluated using SDS-PAGE gel electrophoresis with 2 mg/mL WCP.

#### 4.3.2. Assays of UV Spectroscopy and Molecular Weight

A UV–visible spectrophotometer (SP-756P, Shanghai Spectrum Instruments Co., Ltd, Shanghai, China) was used to check the presence of proteins and nucleic acids of WCP. High-performance gel-permeation chromatography (HPGPC) (Shimadzu-GPC-20A, Kyoto, Japan) was used to determine the average molecular weight of WCP (1 mg/mL) with a chromatographic column (TSK-gel GMPWXL column, TOSOH, Tokyo, Japan). The experimental conditions were as follows: temperature of column, 35 °C; temperature of detector, 35 °C; injection volume, 20 μL; mobile phase, 0.1 N NaNO_3_ and 0.06% NaN_3_ water solution; flow rate, 0.6 mL/min. The standard of Pullulan (Narrow MWD) was used to establish the standard curve, and the standard curve was used to calculate the average molecular weight of WCP [52].

#### 4.3.3. Scanning Electron Microscopy

The morphological characteristics of WCP were observed with a scanning electron microscope (Axio Imagerm2 EVO10, ZEISS, Oberkochen, Germany) under the condition of accelerating voltage (6 kV). Freeze-dried WCP was fixed with double-sided tape, sputtered with gold, and then scanned under high-vacuum conditions [53].

#### 4.3.4. Assays of FT-IR Spectrum and Monosaccharide Composition

WCP (1 mg) and potassium bromide (150 mg) were blended and pressed into pellets. The pellets were used to detect the existence of functional groups by Fourier transform infrared spectrophotometry (iS10, Nicolet, Waltham, MA, USA) in a region spanning 4000–400 cm^−1^ [54].

Liquid chromatography (LC) (LC-20AD, Shimadzu, Kyoto, Japan) was used to determine the monosaccharide composition. WCP was hydrolyzed into monomers with 2 M trifluoroacetic acid (120 °C, 4 h), and dried with nitrogen. The hydrolysates were dissolved in 3 mL of ultrapure water. Precisely 250 µL of the hydrolyzed sample solution was aspirated into a 5 mL EP tube, then 250 µL of 0.6 M NaOH and 500 µL of 0.4 M PMP-methanol were added and reacted at 70 °C for 1 h. The mixture was cooled in cold water for 10 min, then 500 µL of 0.3 M HCl was added to neutralize it, followed by the addition of 1 mL chloroform and vortexing for 1 min. Centrifugation was carried out for 10 min at 3000 rpm. The supernatant was obtained for analysis. The analytical conditions of the LC were as follows: detector, SPD-20A; column temperature, 30 °C; analytical column, Xtimate C18 column (200 mm × 4.6 mm); mobile phase, potassium dihydrogen phosphate (50 mM) and acetonitrile; injection volume, 20 μL; flow rate, 1.0 mL/min. Mannose, ribose, rhamnose, glucuronic acid, galacturonic acid, *N*-acetyl-aminoglucose, glucose, *N*-acetyl-aminogalactose, galactose, xylose, arabinose, and fucose were chosen as references.

### 4.4. Antitumor Activity In Vivo

#### 4.4.1. Animal Experimental Design

Thirty male BALB/c mice (7–8 weeks old, body weight 20 ± 2 g) were bought from the SPF (Beijing, China) Biotechnology Co., Ltd. (license number: SCXK Beijing 2019–0010). The entire process of animal experimentation complies with the National Institute of Health’s guidelines for care [55], which was approved by the Ethics Committee of Experimental Animal Care at Chengdu University of Traditional Chinese Medicine (No. 2022DL-019). The animals were kept in an SPF room with a relative humidity of 40–60%, temperature of 20–22 °C, and a light–dark cycle of 12 h (license number: SYXK-2020-124). They were fed freely with water and food. After one week of acclimatization, the mice were divided into five groups randomly, namely the NC group, Model group, 5-FU group, Low WCP group, and High WCP group. Mice in the NC and Model groups received saline gavage in a volume of 0.2 mL per day. WCP groups were intragastrically treated with an equal volume of WCP at a dose of 100 mg/kg and 300 mg/kg as a reference, respectively [50]. After 7 days of gavage, the tumor-bearing mice model was established by injecting 0.2 mL of H22 hepatoma cell suspension (1 × 10^7^ cells/mL) into the axilla of the left forelimb [56]. Mice were treated continuously with saline and WCP for 3 weeks. 5-fluorouracil (20 mg/kg) was injected into the mice of the 5-FU group once every day. During the experiments, the mice were weighed, observed, and had their physical characteristics recorded every day.

#### 4.4.2. Solid Tumors and Immune Organ Indices

After the experiment, mice were anaesthetized with pentobarbital and then executed. The blood samples, solid tumors, liver, spleen, lung, and thymus were collected and weighed. The organ indices were calculated by dividing the organ weight (g) by the body weight (g) [39]. The tumor size was measured by vernier caliper and the tumor volumes were calculated by tumor length × width^2^ by 2 [57].

#### 4.4.3. Blood Routine Examination and Blood Biochemical Analysis

The fresh blood (250 μL) samples were added to 0.5 mL K_3_ ethylenediaminetetraacetic acid (K_3_EDTA)-coated tubes. After mixing, the samples were analyzed by BC-5120 automatic blood analyzer (Mindray, Shenzhen, China). The levels of serum ALT and AST were measured by BS-240 VET automatic biochemical analyzer (Shenzhen Mindray Biomedical Electronics Co., Ltd., Shenzhen, China).

#### 4.4.4. TUNEL Assay

A One-step TUNEL In Situ Apoptosis Kit (Green, FITC) (Elabscience, Wuhan, China) was applied to detect DNA degradation in cells according to the manufacturer’s instructions. Briefly, slides were soaked in Citrisolv solution I and Citrisolv solution II for 15 min, respectively. The slides were then washed with a range of concentrations of ethanol (100%, 95%, 85%, 75%, and 0%), each for 5 min. These slides were dried slightly and treated with proteinase K^+^. Then, the TDT Equilibration Buffer was added to the sections and incubated at 37 °C for 30 min. The TUNEL reaction mixture was added to the sections and incubated in a humidified chamber in the dark at 37 °C for 1 h. After washing three times with PBS, the slides were stained using DAPI (4′,6-diamidino-2-phenylindole) (Servicebio, Wuhan, China). Finally, the stained sections were observed under a Panoramic SCAN II (3DHISTECH Kft, Budapest, Hungary) [58].

#### 4.4.5. T Lymphocyte Subsets, Macrophage Distribution and Proportions

Firstly, the freshly obtained spleen was ground, then the grinding solution was transferred to a 15 mL centrifuge tube and centrifuged at 1300 rpm for 5 min at 4 °C; the supernatant was discarded and the bottom of the tube was tapped on the table to disperse the cells. Then, 4 mL of erythrocyte lysate was added, the mixture was left to stand for 5 min, and 4 mL of 1640-FBS was added to terminate the reaction, followed by centrifugation at 4 °C for 5 min at 1300 rpm. Finally, 2 mL of 1640-10% FBS was added to the whole mouse spleen, with a cell count of approximately 5 × 10^7^ cells/mL. The distribution and proportions of T lymphocytes subsets and macrophages in the spleen were evaluated using flow cytometry (FACSCallibur, BD, Franklin Lakes, NJ, USA). Monoclonal antibodies of FITC-conjugated CD4, PE-conjugated CD8, and PE-conjugated F4/80 were applied to stain these spleen cells, which were detected and analyzed using FlowJo_V10 [59,60].

#### 4.4.6. Evaluation of Cytokine Levels and Lungs Fixed with Bounis Fixative

The levels of IL-10 were determined using the corresponding ELISA kit, and the result was calculated following the manufacturer’s instructions. The lung tissue obtained from the tumor-bearing mice was washed twice with PBS and submersed in Bounis for staining overnight [61].

#### 4.4.7. HE Assay of Liver and Tumor

The HE staining of the liver and tumor of H22 tumor-bearing mice was conducted following the manufacturer’s instructions, and the results were obtained by optical microscopy. The HE staining results of the liver and tumor were shown as color images at 200× and 400× magnification [62].

#### 4.4.8. Western Blot Assay and Quantitative Real-Time PCR

Western blotting is used for the qualitative and quantitative analysis of proteins [63]; the total protein in tumor tissues was extracted and measured by a BCA assay kit (Beyotime, Shanghai, China). All proteins were separated using 10% SDS-denaturing polyacrylamide gel and then transferred to PVDF membranes. The membranes were incubated with antibodies against STAT3 (1:1000), p-STAT3 (1:1000), Bax (1:1000), Caspase3 (1:1000), Caspase8 (1:1000), BCl-2 (1:1000), and Cyto-c (1:1000) overnight at 4 °C. The membranes were then washed in TBST and incubated with horseradish peroxidase-conjugated sheep anti-rabbit (1:8000) at room temperature for 1 h. Finally, the enhanced chemiluminescence (ECL) reagent was used to observe the proteins with a BIO-RAD Gel Doc™ XR+ Imaging System (Bio-Rad, Hercules, CA, USA).

The method of qRT-PCR was carried out as described in the literature [64]. The total RNA was isolated from the tumor tissue using Trizol reagent, and reverse transcribed into cDNA. A real-time SYBR Green PCR reagent was used to detect the cDNA sample with specific primers (Table 2). The mRNA expression levels of Bcl-2, Bax, IL-6, IL-Iβ, NF-κB, TNF-α, and GAPDH were calculated by the 2^−ΔΔct^ method and expressed as a ratio compared to the model.

### 4.5. Statistical Analysis

At least three independent trials were conducted in all experiments, and the data were expressed as mean ± standard deviation. The results were analyzed using the GraphPad Prism software (version 9.0.0; GraphPad Inc., La Jolla, CA, USA), and the statistical significance of differences (*p* < 0.05) was evaluated using ANOVA multiple comparison tests.

## 5. Conclusions

In this study, high-molecular-weight WCP was prepared and its activity in H22 tumor-bearing mice was investigated. Our results demonstrated that WCP treatment could promote the protection of the immune organs, activate the activities of immune cells, and decrease the expression of the IL-10 of H22 tumor-bearing mice. In addition, in this study, WCP was discovered to inhibit hepatocellular carcinoma, possibly by promoting apoptosis through the promotion of Cyto-c/Caspase8/3 and inhibition of IL-10/STAT3/Bcl2 signaling pathways, which will advance our understanding of WCP as a potential therapeutic option for HCC.

## Figures and Tables

**Figure 1 molecules-28-03351-f001:**
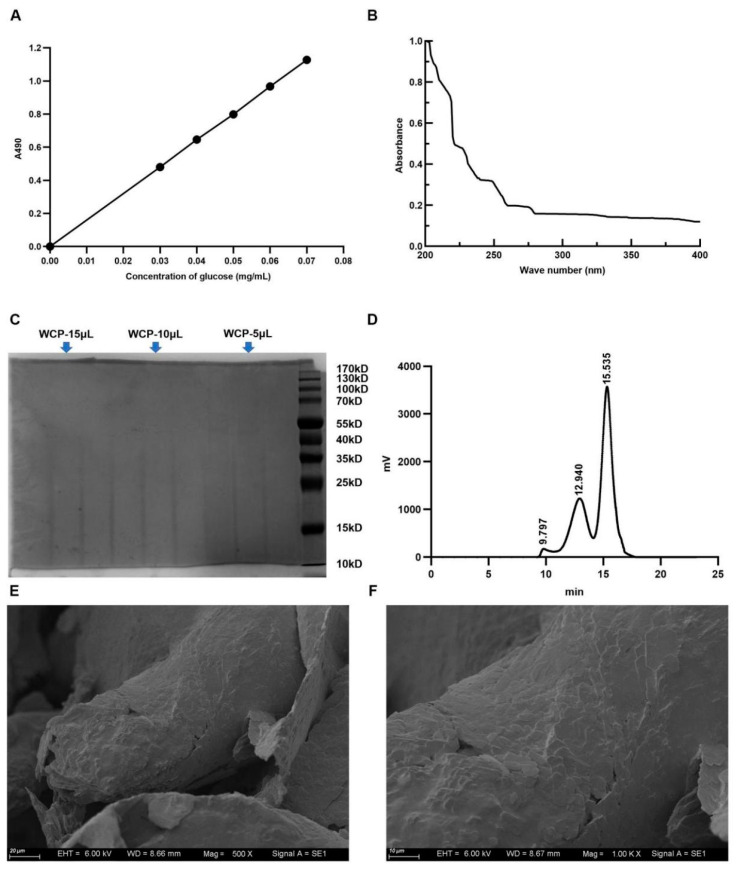
The total sugar content, purity, average molecular weight, and microstructural features of WCP. (**A**) The total sugar content in the WCP was measured by phenol-sulfuric acid method, using glucose as a standard control. (**B**,**C**) The purity of WCP was determined by UV spectrum and SDS-PAGE gel electrophoresis. (**D**) The average molecular weight of WCP was measured by HPGPC. (**E**,**F**) The microstructural features of WCP were observed by scanning electron microscopy (500×, 1000×, scale bar = 20 µm and 10 µm, respectively).

**Figure 2 molecules-28-03351-f002:**
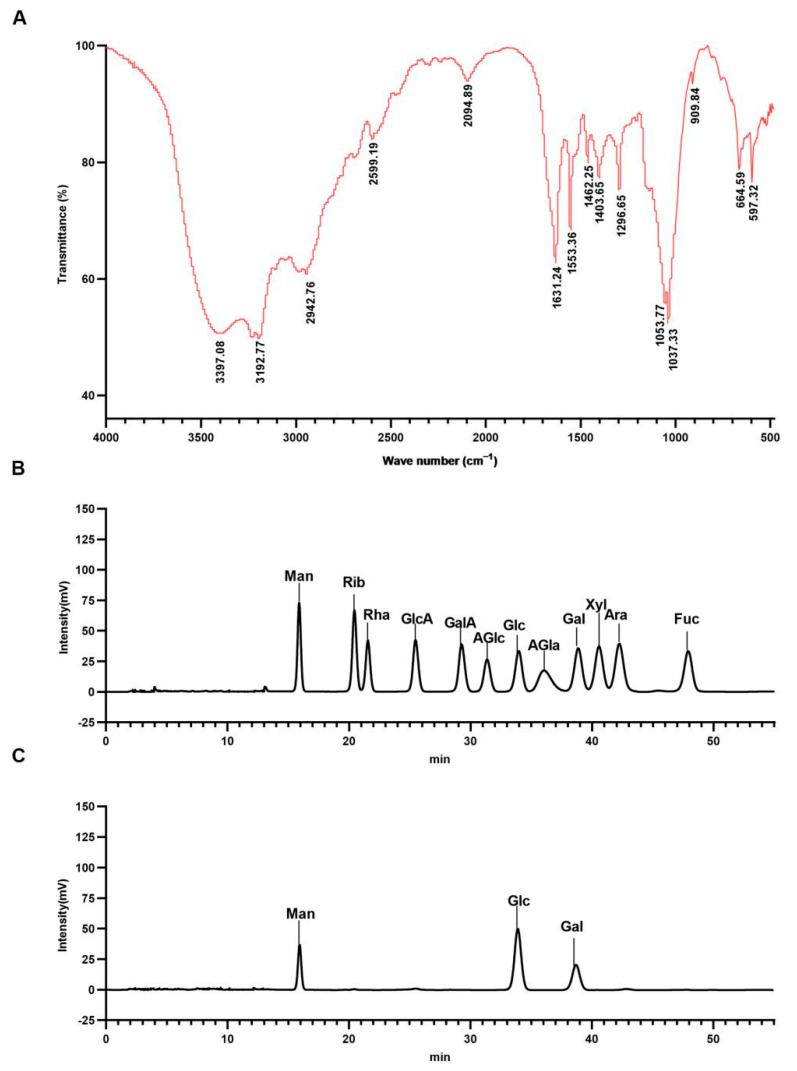
Main organic functional groups and monosaccharide composition of WCP. (**A**) The main organic functional groups in WCP were detected by the FT-IR spectrum. (**B**,**C**) LC analysis of standard monosaccharides and WCP. Man (mannose); Rib (ribose); Rha (rhamnose); GlcA (glucuronic acid); GalA (galacturonic acid); AGlc (n-acetyl-d-glucosamine); Glc (glucose); AGla (n-acetamidogalactose); Gal (galactose); Xyl (xylose); Ara (arabinose); Fuc (fucose).

**Figure 3 molecules-28-03351-f003:**
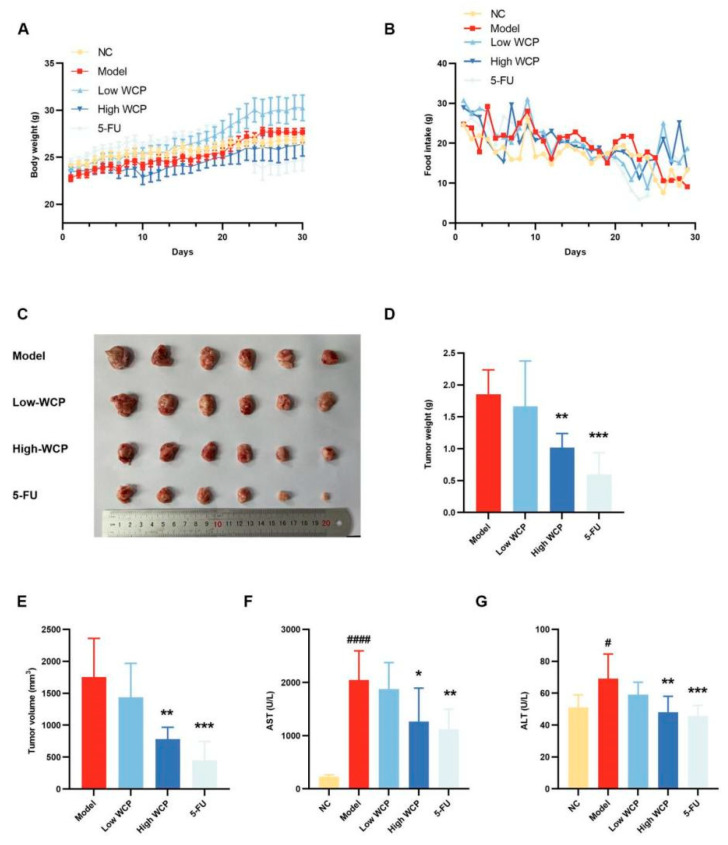
WCP alleviated the symptoms of H22 tumor-bearing mice. (**A**,**B**) Effects of WCP on the body weight and food intake. (**C**) Photographs of tumors from each group. (**D**,**E**) Tumor weight and volume of tumors. (**F**,**G**) Effects of WCP on the levels of serum AST and ALT. ^#^ *p* < 0.05, ^####^ *p* < 0.0001 compared to NC group; * *p* < 0.05, ** *p* < 0.01, *** *p* < 0.001 compared to model group.

**Figure 4 molecules-28-03351-f004:**
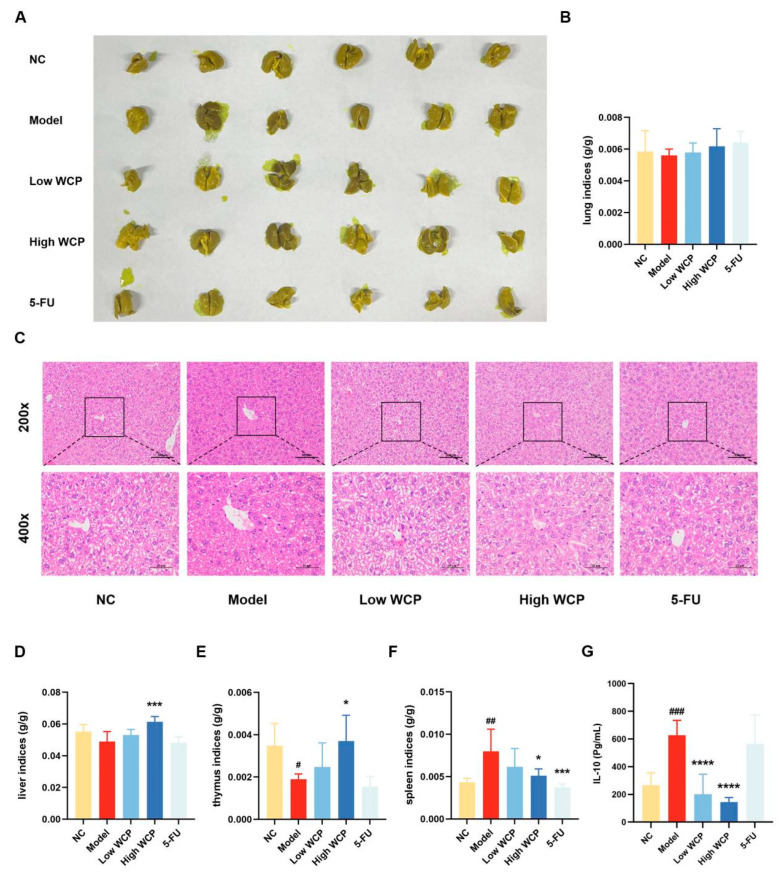
Effect of WCP on the liver, lung, organ indices, and IL-10. (**A**) Lungs fixed with Bouins fixative. (**B**) Lung indices. (**C**) HE staining of liver (200×, 400×, scale bar = 100 µm, 50 µm). (**D**) Liver indices. (**E**) Thymus indices. (**F**) Spleen indices. (**G**) Serum IL-10 concentration of H22-tumor-bearing mice. ^#^ *p* < 0.05, ^##^ *p* < 0.01, ^###^ *p* < 0.001 compared to NC group; * *p* < 0.05, *** *p* < 0.001, **** *p* < 0.0001 compared to model group.

**Figure 5 molecules-28-03351-f005:**
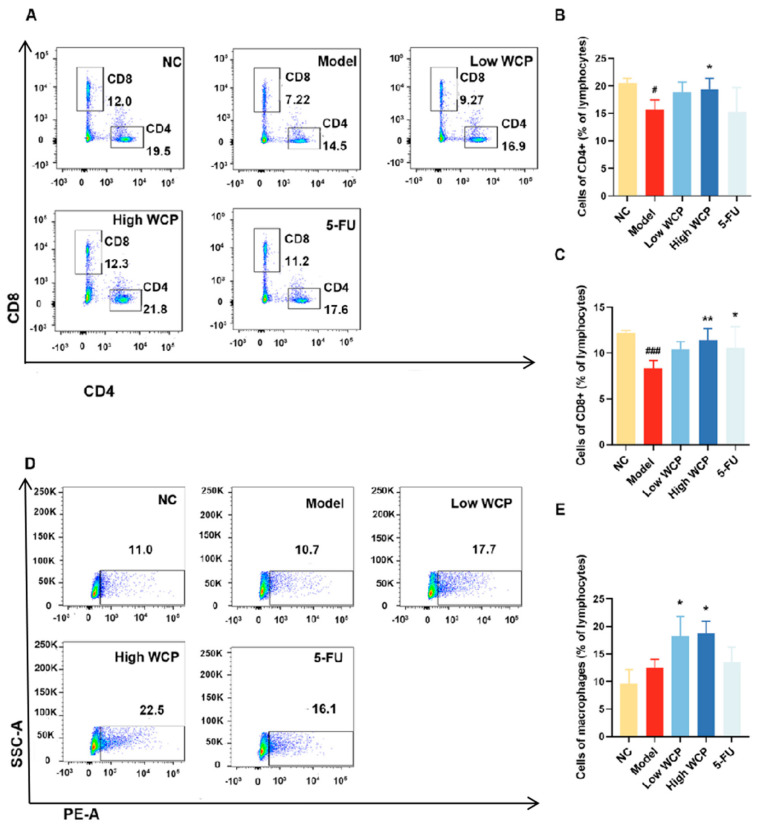
WCP changed the proportions of T lymphocytes and macrophages. (**A**–**C**) FITC/PE staining and flow cytometry analysis showed the proportions of CD4^+^/CD8^+^ in the spleen of H22 tumor-bearing mice. (**D**,**E**) PE staining and flow cytometry analysis showed the proportions of macrophages in the spleen of H22 tumor-bearing mice. ^#^
*p* < 0.05, ^###^
*p* < 0.001 compared to NC group; * *p* < 0.05, ** *p* < 0.01 compared to model group.

**Figure 6 molecules-28-03351-f006:**
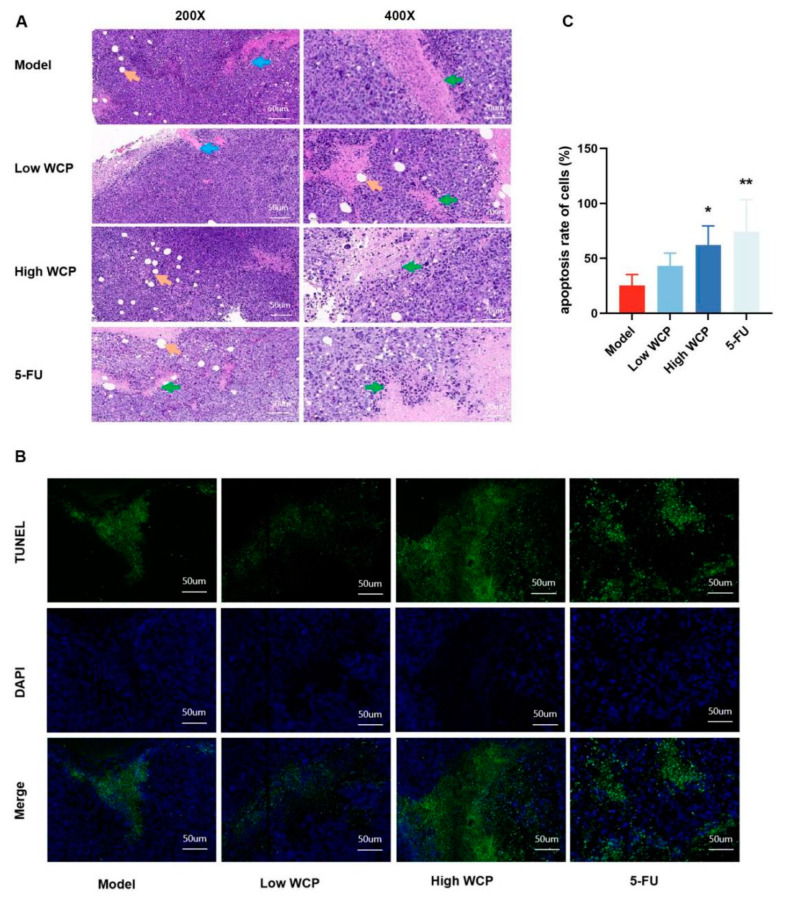
WCP treatment promotes apoptosis of tumor cells. (**A**) HE staining of tumors (200×, 400×), scale bar = 50 µm and 20 µm. Green arrows represent necrotic tissue, orange arrows represent fat vacuoles, and blue arrows represent tumor cells invading muscle tissue. (**B**,**C**) TUNEL detection and proportions of tumor. * *p* < 0.05, ** *p* < 0.01 compared to model group.

**Figure 7 molecules-28-03351-f007:**
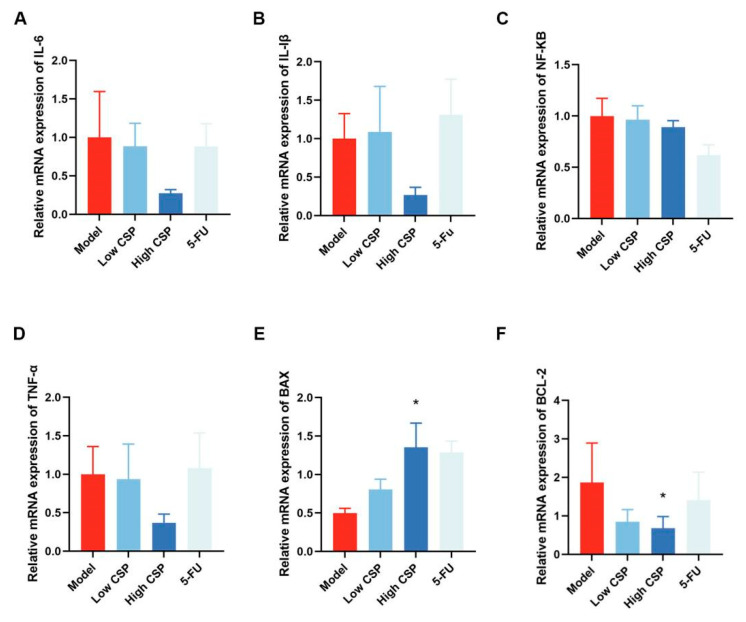
WCP promotes Cyto-c/Caspase8/3 and inhibits IL-10/STAT3/Bcl2 pathway. (**A**–**F**) Relative mRNA expression of IL-6, IL-Iβ, NF-κB, TNF-α, Bax, and Bcl2. * *p* < 0.05 compared to model group.

**Figure 8 molecules-28-03351-f008:**
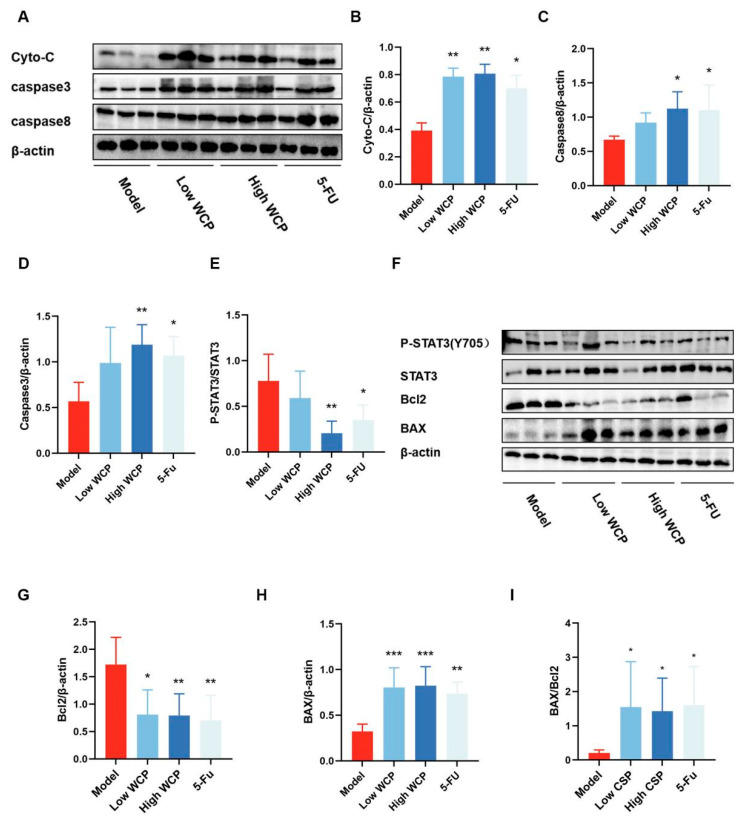
WCP promotes Cyto-c/Caspase8/3 and inhibits IL-10/STAT3/Bcl2pathway. (**A**,**F**) Photographs of the proteins in each group. (**B**–**E**,**G**–**I**) Relative protein expression of Cyto-c, Caspase8, Caspase3, p-STAT3, Bcl2, Bax, and Bax/Bcl2. * *p* < 0.05, ** *p* < 0.01, *** *p* < 0.001 compared to model group.

**Figure 9 molecules-28-03351-f009:**
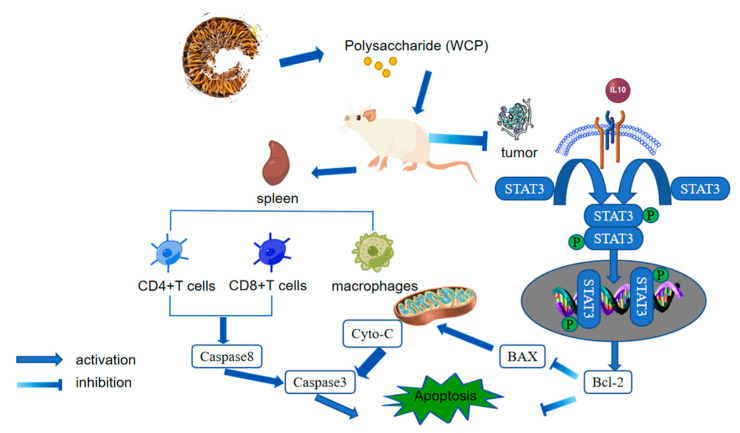
Schematic of wild Cordyceps polysaccharide inhibited the proliferation of H22 cell.

**Table 1 molecules-28-03351-t001:** The blood routine examination of H22 tumor-bearing mice.

Items	Units	Blank	Model	Low WCP	High WCP	5-FU
Leukocytes	10^9^/L	4.49 ± 1.31	8.96 ± 2.04 ^###^	9.81 ± 2.04	8.75 ± 1.43	2.28 ± 0.97 ****
Neutrophils	10^9^/L	0.57 ± 0.27	5.36 ± 1.49 ^###^	5.21 ± 2.70	4.30 ± 1.38	0.17 ± 0.23 ****
Lymphocytes	10^9^/L	4.17 ± 0.85	2.46 ± 0.64 ^#^	3.46 ± 0.65	4.72 ± 1.12 ***	2.10 ± 0.83
Neutrophils percentage	%	15.43 ± 7.19	67.58 ± 9.56 ^####^	54.70 ± 25.05	50.13 ± 20.74	6.15 ± 7.20 ****
Lymphocytes percentage	%	87.47 ± 8.00	40.23 ± 15.26 ^####^	35.17 ± 5.42	43.17 ± 7.92	93.22 ± 7.60 ****
Erythrocytes	10^12^/L	11.91 ± 0.37	10.01 ± 0.76 ^###^	10.01 ± 0.33	10.68 ± 0.89	10.76 ± 0.93
Hemoglobin	g/L	181.17 ± 4.31	151.67 ± 9.42 ^##^	147.00 ± 12.87	155.17 ± 18.31	165.67 ± 10.80
Hematokrit	%	53.13 ± 1.34	45.92 ± 2.36 ^##^	44.13 ± 3.55	45.87 ± 4.30	48.07 ± 3.08
Mean corpuscular volume	fL	44.63 ± 0.91	45.98 ± 1.49	45.78 ± 0.70	44.97 ± 1.27	44.77 ± 1.17
Mean hemoglobin	pg	15.22 ± 0.30	15.17 ± 0.52	15.27 ± 0.10	15.20 ± 0.23	15.40 ± 0.57
Mean hemoglobin concentration	g/L	341.17 ± 4.07	329.83 ± 5.74	333.17 ± 3.19	337.67 ± 8.45	344.67 ± 9.61 ***
Erythrocytes distribution width coefficient	%	15.23 ± 0.37	15.6 ± 0.44	15.72 ± 0.58	15.93 ± 0.62	15.12 ± 0.26
Erythrocytes distribution width standard	fL	28.6 ± 0.73	29.85 ± 0.96	29.92 ± 1.09	30.03 ± 1.47	28.38 ± 0.80
Blood platelet	10^9^/L	828.17 ± 65.14	918.17 ± 111.6	921.67 ± 123.1	899.17 ± 85.21	509.67 ± 242.1
Mean platelet volume	fL	6.22 ± 0.23	6.5 ± 0.34	6.32 ± 0.24	6.25 ± 0.32	6.68 ± 0.26
Platelet distribution width		15.98 ± 0.23	15.97 ± 0.34	15.8 ± 0.35	15.85 ± 0.27	16.05 ± 0.22
Thrombocytocrit	%	0.51 ± 0.02	0.59 ± 0.05	0.58 ± 0.08	0.56 ± 0.06	0.34 ± 0.16 ***
platelet-larger cell count	10^9^/L	74.17 ± 11.82	94.67 ± 16.22	82.83 ± 20.49	81.67 ± 16.06	59.5 ± 26.94 *
platelet-larger cell ratio	%	9.05 ± 1.75	10.55 ± 2.62	9.02 ± 2.12	9.13 ± 1.80	11.75 ± 1.49

^#^ *p* < 0.05, ^##^ *p* < 0.01, ^###^ *p* < 0.001, ^####^ *p* < 0.0001 compared to NC group. * *p* < 0.05, *** *p* < 0.001, **** *p* < 0.0001 compared to model group.

**Table 2 molecules-28-03351-t002:** Primers used in the present study.

Primer	Sequence (5′-3′)
IL-6 F	AATGTCGAGGCTGTGCAGATTAGTAC
IL-6 R	GGGTGGTGGCTTTGTCTGGATTC
Bax-F	GGCGAATTGGAGATGAACTG
Bax-R	AAAGTAGAAGAGGGCAACCA
Bcl-2-F	AGGATTGTGGCCTTCTTTGA
Bcl-2-R	ACCTACCCAGCCTCCGTTAT
IL-Iβ-F	TACTGCCGTCCGATTGAGAC
IL-Iβ-R	TCCAGGGCTTCATCGTTACA
NF-κB-F	TGCGATTCCGCTATAAATGCG
NF-κB-R	ACAAGTTCATGTGGATGAGGC
TNF-α-F	GCACTGAGAGCATGATCCGAGAC
TNF-α-R	CGACCAGGAGGAAGGAGAAGAGG
Gapdh-F	TGTGTCCGTCGTGGATCTGA
Gapdh-R	GATGCCTGCTTCACCACCTT

## Data Availability

The datasets used and/or analysed during the current study are available from the corresponding author on reasonable request.

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
