# Peer review of "The Large Molecular Weight Polysaccharide from Wild Cordyceps and Its Antitumor Activity on H22 Tumor-Bearing Mice"

_molecules, 2023, doi:10.3390/molecules28083351_

Round 1

Reviewer 1 Report

1.        This study mainly discusses the effect and mechanism of C. sinensis polysaccharides on H22 liver cancer. Therefore, it is recommended that the references in the second paragraph of the introduction (15, 16, 17, 19, and 20) be replaced with the literature on the antitumor activity of other traditional Chinese medicine polysaccharides on H22 tumor-bearing mice. In addition, the basis for selecting H22 tumor in this study, or the significance of the investigation, needs to be added.

2.        Line 122, “Sevage” should be written as “Sevag”! This method was derived from Sevag, NOT SEVAGE! (M.G. Sevag, The isolation of the components of streptoeoeeal nueleoproteins in serologieally active form. J. Biol. Chem., 124 (2) (1938): 425-436.).

3.        In section 2.3.4, the description of the monosaccharide composition determination method should be checked. It is absurd that unlabeled sugars are analyzed by the SPD20A ultraviolet detector!

4.        The UV spectrum of WCP is unqualified and needs to be rescanned.

5.        What are the detection limits and quantitative limits for each standard monosaccharide in the analysis of monosaccharide composition? Figure 2C should be improved, and the absorption peaks for monosaccharides that exceed the detection limit should be shown, such as Rha, GlcA, GalA, Ara and Fuc.

Author Response

Replies to reviewer’s comments

Thank you very much for your useful comments and suggestion concerning our manuscript entitled " Structural characterization of polysaccharide from Wild Cordyceps and its antitumor activity on H22 tumor-bearing mice "(ID 2319906). We have carefully reviewed the comments and have revised the manuscript accordingly. Our responses are given in a point-by-point manner below. Changes to the manuscript are shown in red. We hope that the revised version would meet the requirements of your journal and could be accepted for publication.

Thank you again.

Yours sincerely,

Li Tan

Yanan Hua (Ph.D.) and Jinling Guo (Pfo.)

Chengdu University of Traditional Chinese Medicine

CHINA

E-mail: guo596@cdutcm.edu.cn

Reviewer  

Q1: This study mainly discusses the effect and mechanism of C. sinensis polysaccharides on H22 liver cancer. Therefore, it is recommended that the references in the second paragraph of the introduction (15, 16, 17, 19, and 20) be replaced with the literature on the antitumor activity of other traditional Chinese medicine polysaccharides on H22 tumor-bearing mice. In addition, the basis for selecting H22 tumor in this study, or the significance of the investigation, needs to be added.

Reply: Thank you very much for your valuable suggestions concerning our manuscript. References (15, 16, 17, 19, 20) in the second paragraph of the introduction have been replaced by literature on the antitumor activity of other Chinese herbal polysaccharides in mice with H22 tumors. The selection of H22 tumors as the disease model for this study was based on the fact that there is no clarity on the antitumor active component of Cordyceps, and the anti-H22 tumor activity of polysaccharides obtained from mycelia of fungi (Cordyceps sinensis) isolated from wild Cordyceps after artificial culture was reported[1]. Considering that there is a certain similarity between the functions of the two kinds of polysaccharides [2], and the molecular weight of Cordyceps polysaccharides is much larger than that of mycelial polysaccharides (some studies have demonstrated that polysaccharides with larger molecular weight had better anticancer activity [3]), we assume that Cordyceps polysaccharide has better inhibitory effect on hepatocellular carcinoma.

Reference

[1]Yang ML, Kuo PC, Hwang TL, Wu TS. Anti-inflammatory principles from Cordyceps sinensis. J Nat Prod. 2011 Sep 23;74(9):1996-2000. doi: 10.1021/np100902f. Epub 2011 Aug 17. PMID: 21848266.

[2]Wang J, Nie S, Kan L, Chen H, Cui SW, Phillips AO, Phillips GO, Xie M. Comparison of structural features and antioxidant activity of polysaccharides from natural and cultured Cordyceps sinensis. Food Sci Biotechnol. 2017 Feb 28;26(1):55-62. doi: 10.1007/s10068-017-0008-3. PMID: 30263510; PMCID: PMC6049471.

[3]Zhou X, Gong Z, Su Y, Lin J, Tang K. Cordyceps fungi: natural products, pharmacological functions and developmental products. J Pharm Pharmacol. 2009 Mar;61(3):279-91. doi: 10.1211/jpp/61.03.0002. PMID: 19222900.

Q2:Line 122, “Sevage” should be written as “Sevag”! This method was derived from Sevag, NOT SEVAGE! (M.G. Sevag, The isolation of the components of streptoeoeeal nueleoproteins in serologieally active form. J. Biol. Chem., 124 (2) (1938): 425-436.).

Reply: We are very sorry for our mistake. Sevage has been amended to Sevag in line 125,marked in red. Thank you for your valuable suggestion.

Q3:In section 2.3.4, the description of the monosaccharide composition determination method should be checked. It is absurd that unlabeled sugars are analyzed by the SPD20A ultraviolet detector!

Reply: Thanks for your advice. Cordyceps polysaccharides were analyzed after hydrolysis followed by PMP-methanol derivatization, which has been modified in the method section. Modified section in Line 157-162, marked in red.

Q4:The UV spectrum of WCP is unqualified and needs to be rescanned.

Reply: The UV spectrum of the WCP has been rescanned. Modified section in Line 276, Figure 1 B. 

Q5What are the detection limits and quantitative limits for each standard monosaccharide in the analysis of monosaccharide composition? Figure 2C should be improved, and the absorption peaks for monosaccharides that exceed the detection limit should be shown, such as Rha, GlcA, GalA, Ara and Fuc.

Reply: In the section on monosaccharide composition, the limit of detection for each standard monosaccharide is 1µg/mL and the limit of quantification is 3µg/mL, Figure 2C has been improved to show the absorption peaks of monosaccharides above the limit of detection. Modified section in Line 294, marked in red and Line 295, Figure 2C.

Reviewer 2 Report

The manuscript entitled (Structural characterization of polysaccharide from Wild Cordyceps and its antitumor activity on H22 tumor-bearing mice) is an interesting and well valued.  However, there are few raised points that required more attention by the authors before being accepted for publication.

Ø  the abstract needs rephrasing to be more clear emphasizing the results.

Ø  line 66 showed exhibited delete one word same meaning.

Ø  check all genus and species to be italics.

Ø  figure 1C, E AND F need to be more clear.

Ø  polysaccharide is characterized only by gel, gel electrophoresis , the authors have to mention the reason for the selection of this technique and if there are more techniques that can be applied

Ø  The manuscript needs a substantial rewriting and rewording, particular attention to the use of English (style, grammar and spelling). A native speaking English should be check the whole manuscript.

Ø   

Author Response

Replies to reviewer’s comments

Thank you very much for your useful comments and suggestion concerning our manuscript entitled " Structural characterization of polysaccharide from Wild Cordyceps and its antitumor activity on H22 tumor-bearing mice "(ID 2319906). We have carefully reviewed the comments and have revised the manuscript accordingly. Our responses are given in a point-by-point manner below. Changes to the manuscript are shown in red. We hope that the revised version would meet the requirements of your journal and could be accepted for publication.

Thank you again.

Yours sincerely,

Li Tan

Yanan Hua (Ph.D.) and Jinling Guo (Pfo.)

Chengdu University of Traditional Chinese Medicine

CHINA

  • mail: guo596@edu.cn

Reviewer 

 Q1:the abstract needs rephrasing to be more clear emphasizing the results.

Reply: Thank you very much for your valuable suggestions concerning our manuscript. The abstract has been reworded. Modified section in Line 26-32, marked in red.

 Q2line 66 showed exhibited delete one word same meaning.

Reply: Thanks for your suggestion. Line 66 descriptions have been confirmed. Modified section in Line 69, marked in red.

 Q3check all genus and species to be italics.

Reply: Thanks for your advice. The issue of italicizing the species in the full text has been confirmed and revised. Modified section in Line 66-77, marked in red.

Q4: figure 1C, E AND F need to be more clear.

Reply: Thanks for your advice. figure 1C is the original image taken on the instrument, the marker is clearly visible, but the resultant image does not show a clear band because the sample is free of protein impurities. Figure 1 E and F are the original image from the instrument has been exported with maximum clarity.

Q5:polysaccharide is characterized only by gel, gel electrophoresis , the authors have to mention the reason for the selection of this technique and if there are more techniques that can be applied

Reply: Thanks for your advice. Modified section in Line 471-475, marked in red.

Q6:The manuscript needs a substantial rewriting and rewording, particular attention to the use of English (style, grammar and spelling). A native speaking English should be check the whole manuscript.

Reply: The article has been rewritted. Modified section marked in blue, english-edited-64066.